# How to Fight Puppy Mills: Toughening the Sentences for Animal Abuse in the Post-Communist Region

**DOI:** 10.3390/ani10061020

**Published:** 2020-06-11

**Authors:** Lukáš Novotný

**Affiliations:** Regional Development Department, The MIAS School of Business, Czech Technical University in Prague, 16000 Prague, Czech Republic; lukas.novotny@cvut.cz; Tel.: +420-22435-3157

**Keywords:** animal abuse, puppy mills, legislation, criminal law, post-communism, Czechia

## Abstract

**Simple Summary:**

The study examines the main legislative issues of providing a legal solution to the problem of illegal puppy mills in the post-communist context. These issues are demonstrated using the Czech Republic, a country that has become infamous for its illegal breeding establishments and subsequent export of puppies and kittens to other European countries, as an example. The country recently adopted tougher sentencing guidelines for animal abuse. The analysis identified three main obstacles to adopting tougher legislation: unwillingness to admit the gravity of the problem of animal abuse and deficient puppy mills; a conservative approach to legislation; inconsistencies caused by the Criminal Code amendment, especially violation of the ultima ratio principle. This was emphasised by a number of criminal law experts, who even warned that the Criminal Code amendment passed would not function in practice. The study demonstrates this on an analysis of criminal law experts’ positions and on the debates that took place in both chambers of the Czech parliament.

**Abstract:**

This study seeks answers to questions regarding the kind of main legislative issues and obstacles there are in providing a legal solution to the problem of illegal puppy mills in the post-communist context, how criminal law experts opine about toughening the sentencing guidelines for animal abuse and deficient puppy mills, what kind of arguments have been formulated and how they have shaped the decision making by lawmakers, and how Czech politicians have argued in favour of or against toughening the sentencing guidelines for animal abuse. The Czech Republic was selected as a country of “flourishing” illegal breeding establishments and puppy exports to other European countries—a problem that has long required a solution. The introduction defines the concepts of animal abuse and puppy mills employed in the paper. Subsequently, the paper outlines existing laws as well as the amendments to toughen the sentencing guidelines. I use the example of debates among parliamentarians and legal experts on toughening the Czech Criminal Code and introducing longer prison terms to demonstrate some typical issues of the debates on tougher sentences for animal abuse in the post-communist region.

## 1. Introduction

Issues of animal abuse and the poor conditions in illegal breeding establishments for dogs, as well as other animals, can be studied from different perspectives: veterinary medicine, animal science, public health, psychology, sociology, political science or law [1]. Legal and political perspectives dominate the present paper. No matter the viewpoint taken, these actions should be condemned by society as well as law. Indeed, humane treatment of nature and animals is one of the qualities of an advanced society [2].

At the same time, this illegal business is especially “flourishing” in the context of post-communist countries. The Czech Republic, along with Slovakia and some other Eastern European countries, serves as the main supplier of dogs and cats across Europe, as consistently reported by the charity Animal Eye [3]. The fact that these countries are faced with a prosperous and often illegal trade in pets is due to the lack of effective legislation to curb this form of serious crime, along with the reluctance to acknowledge it. According to charities, traffickers earn large sums of money every month. This is because puppies, but also kittens and other animals, are born into atrocious conditions, and it is not uncommon for them to die soon thereafter. Although such trafficking is a criminal offence under Czech law, it is extremely difficult to prove. Traditionally, this situation, along with the conditions in the country’s illegal mass puppy mills, have been highlighted primarily by international organisations; fraudulent practices in the dog breeding market have also been mapped by the media, both Czech and international [4]; in the country itself, 37,000 people signed a petition in 2019 calling on politicians to take effective legislative action against animal abuse [5,6]. Although the Czech Republic is known as a country of pet owners, it lacks any generally applicable and binding provisions on the rights and obligations of noncommercial pet breeders. At the same time, commercial breeding falls into the least-regulated category of “unqualified trades”. Furthermore, administrative authorities in the Czech Republic have an extremely limited repertoire of instruments and sanctions vis-à-vis irresponsible or lawbreaking breeders. In contrast, countries like Australia prescribe high pecuniary penalties against commercial puppy mills, and even prison sentences in extreme cases [7].

This study focuses on issues of puppy mills and criminal law protections against animal abuse. As stated above, the perspective it takes is primarily a legal and political one. The issue of animal cruelty and the legal solution to it has received little attention both in the country and on a more general level in the central European area. In addition, previous analyses of parliamentary debates in the Czech Republic have not gone as in depth as in this case. For this study, I opted for a qualitative content analysis of legislation and parliamentary debates. After formulating research questions, one gathers useful material—here debates among criminal law experts and parliamentary debates. Subsequently, one divides the material into coding units depending on the contents of interest, verifies and evaluates the coding framework, and makes any necessary modifications to the framework. Then, one proceeds to the analysis itself and to interpreting the results [8,9,10,11]. The study’s research questions are defined as follows:

RQ1: What are the main legislative issues of and obstacles to providing a legal solution to the problem of illegal puppy mills in the context of post-communist countries?

RQ2: How did criminal law experts opine about toughening the penalties against animal abuse and deficient puppy mills, what kind of arguments were formulated, and how did they shape the subsequent decision making by lawmakers?

RQ3: What were the arguments of Czech politicians in favour of or against toughening the sentencing guidelines for animal abuse?

## 2. Theoretical Background

### 2.1. Animal Abuse

Animal abuse is an umbrella term for human actions which cause intended or negligent harm to animals, actions detrimental to their health, living conditions, physiological wellbeing, ability to live a full life, etc [12]. Different perspectives on animal abuse exist in different population groups, and people’s evaluations of the ways animals are treated depend on specific context and culture. In the predominant view, animal abuse refers to processes that may result in permanent damage to the animal’s health or ultimately in its death, whether due to assault or malpractice [13,14]. Sadly, animals are treated as movable assets in the legal systems of many countries. On the other hand, their protection as living beings is gaining increasing traction [15,16]. This reflects the influence of a more “humane” approach to animals in the legal provisions of Western countries, especially Germany, Austria, Switzerland and France. There has been a gradual shift in our understanding of animals from mere things to living creatures endowed with senses and perception of their emotional bond to others, especially their masters [17].

Animal abuse is becoming a criminal offence for which perpetrators face prison sentences of different length, depending on national legislation [18]. In some countries, it is treated as an administrative offence where the offender must only pay damages or a fine. The legal perspectives on this kind of violations against animals depend on the cultural context and the struggle of local activists [19] to introduce animal rights [20]. Animal rights, or the legal implications of animal abuse, also depend on legislative definitions and the categories of animals covered by animal protection laws. Czech law, for example, protects vertebrate animals in their postembryonic stages.

There is sociological evidence of a relationship between crimes against animals and crimes against human victims [21]. Animal abuse may indicate (and often indeed indicates) future delinquent behaviour. For example, an analysis of 354 cases of serial killers revealed that 75 of them (over 21%) had a pre-existing history of animal abuse [22]. Political debates, too, often see this line of argument in favour of toughening the penalties for animal abuse, as will be shown in the chapters below.

Thus, such forms of violence may serve as warning signs of other abuse, and the discovery of this relationship has important implications for domestic violence prevention and intervention strategies. For that reason, some organisations (e.g., the American Humane Association) strongly emphasise the need for mandatory reporting of animal abuse [23].

### 2.2. Puppy Mills

Above all, breeding establishments are nowadays difficult to detect, thus representing a critical point for civic sector activities. They comprise a relatively well-structured industry [24]. Post-communist countries provide a relatively safe environment for this business: their authorities do not have prosecuting experience and they exist in a kind of post-communist void with no settled idea about the value foundation of human and other rights [25,26]. Strictly speaking, a puppy mill for dogs, cats or other animals is merely a media construct and, until recently, it remained unknown to the legal system of the Czech Republic. This was changed by an amendment to the Act on the Protection of Animals Against Cruelty.

Section 7a, Protection of dogs and cats in their reproduction

‘(1) It is prohibited to breed dogs or cats, including their reproduction, in an establishment, including a dwelling, where dogs or cats are bred and reproduced in inadequate conditions that give rise to their suffering and in larger groups where they cannot satisfy their physiological, biological or ethological needs (hereinafter as puppy mill). A puppy mill is defined as an establishment, including a dwelling, referred to in the first sentence even if it is not the main object of the breeder’s activity to reproduce animals or make profit.’[27]

In legal provisions of other countries of the former communist bloc, protection of animals against cruelty is foreseen as well (see, for example, Sections 378 and 378a of the Criminal Code of the Slovak Republic or Article 4 of the Polish Animal Protection Act). In Slovakia, it is also possible to suspend the execution of a sentence for cruelty to animals.

Scholarly literature, too, refers rather vaguely to such methods of breeding dogs and other animals, whether for business or noncommercial purposes, that are accompanied by certain irregular phenomena. Those include, above all, malpractice in animal care [28], as well as tax evasion in the case of commercial puppy mills. Commercial breeding business is directly driven by public demand, often for high-priced fashionable purebreds. The goal is to maximise revenue while minimising expenditure. According to international organisations and charities, puppy trafficking is the third most lucrative illicit market in Europe, right after drug and arms trafficking [4,29,30]. In these conditions, dogs (or other animals) are subjected to cruel and agonising treatment.

In deficient puppy mills, brood mothers represent merely a source of income; commercial breeders induce mating in every heat cycle and the brood mothers keep reproducing for as long as they are physically capable [31,32]. Breeders often choose the stud to mate with their bitch irrespective of their respective health, pedigree, character or exterior traits. In the absence of checks, breeders often organise the mating of closely related or sick individuals. As a result, many puppies suffer from diseases or birth defects. Traffickers attempt to conceal this with false excuses and give price discounts for ill health. Such discounts, however, will be more than outweighed by the cost of veterinary care. Expired vaccine shots and forged vaccination certificates are no exception. All that is perfectly concealed and difficult to prove. Commercial breeders often keep their animals in inadequately sized cages or pens (many a time puddling in their own faeces), do not provide enough water or feed, and fail to secure the necessary veterinary care [33]. Brood mothers that are no longer capable of reproducing tend to be put down immediately. Puppies are separated from their mothers too soon (typically at the age of three or four weeks), which affects their physiological and mental health.

## 3. Materials and Methods

### 3.1. Research Background

#### 3.1.1. The Czech Republic and the Breeding Business: A Specific Form of Animal Abuse

The breeding business constitutes one of the gravest forms of animal abuse. To blame are the animals’ atrocious living conditions as well as the organised character of the business and related tax evasion in the order of tens, perhaps even hundreds of millions of euros [34]. Traffickers buy out puppies from domestic mills, above all, and offer them abroad for hundreds of euros apiece. Their profits remain untaxed. For example, approximately 30,000 Czech and Slovak puppies are exported to Italy alone. According to journalists, traffickers purchase a puppy in the Czech Republic for no more than one hundred euros to sell it for a thousand in the south. At the same time, Italy is not the top importer of Czech puppies—larger numbers are sold to Germany or Belgium [34].

This is a highly problematic area of animal abuse. Deficient puppy mills keep dogs, as well as other animals, in isolation from people, thus preventing the necessary socialisation of young ones. Some puppies are sold from trafficker to trafficker before they arrive at their final owner. Yet, only then do they often die due to an incomplete immunisation process.

For these reasons, the demand to increase the prison terms for the gravest forms of animal abuse has long been voiced by the expert community and laypeople in the Czech Republic (a petition was signed by more than 37,000) as well as by many criminal judges. The international dimension of this crime is of special relevance here. One reason the Czech Republic and Slovakia have been nicknamed “Europe’s puppy mills” is that, according to expert estimates, up to 70% of all animals born there are exported. Such illegally bred animals are never registered, and the related incomes are almost never taxed [24].

In this organised, hidden and highly profitable illegal activity, the sale prices can be in the order of thousands of euros per exemplar. Lawyers and judges argue that the existing administrative sanctions (with fines averaging EUR 400–800) are ineffective or at least out of proportion to how lucrative this business is. Moreover, the illegal activities are typically conducted in complete secrecy. The animals are often kept in soundproof underground premises, the perpetrators frequently change their SIM cards and use false identities, and dogs are transferred without signing contracts or exchanging contact information [27].

#### 3.1.2. Czech Animal Protection Laws

Generally speaking, existing legal provisions allow any natural person to pursue animal husbandry, and the number of animals kept by a person is practically unlimited as long as adequate care is provided. The pet keeping conditions prescribed are so vague that inspection authorities cannot rely on a robust legal basis and clear interpretation of the law. The Act on the Protection of Animals Against Cruelty [35] requires pet keepers to secure adequate conditions for preserving a pet’s physiological functions and satisfying its biological needs so that no pain or harm to health is inflicted, and to take measures to prevent animals from escaping captivity. Pet keeping is considered illegal if any these conditions are violated or if the conditions are met but the animal is unable to adapt. Similarly, pet keeping is prohibited if the keeper has created such breeding conditions as to affect the next generations of their animals with genetic disorders such as missing, functionally impaired or mutilated body parts or organs. In principle, though, every pet keeper, or every person who has adopted a stray or abandoned animal, is responsible for the animal’s health and wellbeing; this condition is also effectively met when the finder reports the animal’s whereabouts to the local authority or transfers a stray or abandoned animal to a shelter.

The existing criminal offence of Maltreatment of Animals in the Czech Republic punishes acts of intentional and especially cruel or agonising animal abuse with a prison sentence of up to two years. Stricter sentencing guidelines (up to three years) are prescribed for recidivism or acts resulting in permanent damage to the animal’s health or its death, and even more stringent penalties (up to five years) apply to maltreatment of a “larger amount of animals” (seven or more in accordance with case law) [36]. The current versions of the criminal offences covering animal abuse (the intentional Maltreatment of Animals in Section 302 and the Negligent Omission of Animal Care in Section 303) are categorised in Chapter VIII of the Criminal Code entitled, Criminal Offences against Environment, alongside the offences of Unauthorised Disposing with Protected Wild Animals and Herbs, Poaching, Disposal with Substances Affecting Efficiency of Livestock, etc.

The status and treatment of animals and especially their protection against cruelty are currently regulated by a relatively large body of laws. The approach of Czech private law is based on Section 494 of the Civil Code which ascribes to every animal ‘special significance and value as a living creature endowed with senses’ (Act No. 89/2012 Coll.) The Code provides that ‘a living animal is not a thing’ and prescribes that ‘the provisions on things apply, by analogy, to a living animal only to the extent in which they are not contrary to its nature’. This instruction would probably have to be taken into consideration in cases of acts of simultaneous Maltreatment of Animals and “Damage to a Thing of Another” (compare Section 134 of the Criminal Code for an identical approach). Czech law is inspired by European provisions that have escaped Roman law’s dogma of “mooing instruments”, whereby animals are treated as things [37].

The approach of Czech public law (provisions on obligations and rights that are inspected and sanctioned by the public administration) relies on the definition of animals in the Act on the Protection of Animals Against Cruelty [38]. Here, ‘animals are living beings and are capable of experiencing pain’ and are defined as every ‘live vertebrate, other than man, excluding foetal or embryonic forms’. This definition is of fundamental relevance to specific criminal offences prescribed against animal abuse. The above-mentioned act also defines “cruelty” by providing an unexhaustive list of 25 forms of animal abuse [35]. Consequently, ‘maltreatment in an especially cruel or agonising manner’ equals any of those forms of cruelty but only in their extreme intensity [35]. As such, these provisions form the interpretive basis of reference for the criminal offence of Maltreatment of Animals [36].

Cruelty to animals and operation of deficient puppy mills is facilitated by existing Czech laws, with their insufficient penalties. In spite of a series of cases over the last years, only a handful of perpetrators (fewer than ten) obtained unsuspended prison sentences, and most of those unsuspended prison sentences were given in the context of multiple counts of crime, probation violations etc. Under normal circumstances, it is practically impossible for first-time offenders to face an unsuspended prison sentence for the “mere” crime of Maltreatment of Animals [39]. In the year 2017, the mean amount of pecuniary penalty for Maltreatment of Animals was CZK 20,000 (approximately EUR 800). There has not been a single unsuspended prison sentence for illegal puppy mills. Likewise, there has been no definition of deficient puppy mills in Czech laws (this is being remedied by the amendment to Act on the Protection of Animals Against Cruelty analysed here, with attempts to adopt a related amendment to the Criminal Code).

Some judges in the Czech Republic, too, have called for toughening the sentencing guidelines and amending the Criminal Code, arguing that adequate sentences cannot be imposed without tougher guidelines [40]. There is an extensive network of puppy mills in the Czech Republic. Along with Slovakia and other Eastern European countries, the Czech Republic serves as a transit hub for dog and cat exports to all of Europe. Ministry of Justice statistics for the past three years indicate zero first-time offenders who received unsuspended prison sentences for the offence of Maltreatment of Animals [41]. In other words, not a single person who committed animal abuse for the first time went to prison in the past three years. Indeed, such an outcome is practically impossible under the existing provisions of the Criminal Code.

The statistics also reveal that animal abuse is typically not the perpetrator’s last offence, as individuals sentenced for Maltreatment of Animals go on to commit other types of crime. According to the Ministry of Justice, 22 Maltreatment of Animals offenders committed other criminal offences in 2018, 18 in 2017 and 23 in 2016 [40]. Note that in 2019, the Czech Republic also passed an amendment to Act No 166/1999 Coll., the Veterinary Act, toughening the notification duty of those keeping brood bitches, increasing penalties and foreseeing the introduction of compulsory, universal, permanent, and unique microchipping of dogs above three months of age [42]. However, this regulation once again reminds us of the limits of functional implementation. The practical result has been the impossibility to identify concrete dogs and their owners. Most offences—whether processed by administrative authorities or criminal courts—end in failure to meet the burden of proof. The infamous position of the Czech Republic in this regard is only paralleled by Slovakia, Poland and Finland, which also do not have the above-mentioned systems in place. Not only is there still no dog registry in place, but dogs under the age of three months are also being trafficked. In any case, at least some experts view the recent adoption of pertinent legislation, and the fact that the registry is being prepared, as a positive step forward.

### 3.2. The Method

The Czech Republic has been selected for a case study to demonstrate the different problems that affect existing regulation as well as the proposed new legislation. My methodology is centred upon content analysis, which is viewed as a research tool, procedure, technique, but also as an approach, methodological orientation or conceptual framework [6,7]. Its goal is to comprehend texts, here parliamentary debates on toughening the sentences for animal abuse, in their cultural and social contexts.

I focus on verbatim transcriptions of debates in both chambers of the Czech parliament, as part of a broader discourse. Here I employ an interpretive approach that requires intensive and detailed effort by the student, being capable of disassembling the content into its constituent parts and subsequently assembling it into wholes that are more coherent. There is one risk of this approach that I strive to minimise in the present paper, namely the potentially highly subjective process of formulating one’s conclusions.

In the first section, I rely on legal provisions and relevant literature, the latter mostly international, to outline the concepts with which I operate in this paper. These include primarily animal abuse (or Maltreatment of Animals) and puppy mills. The second section presents the specifics of animal abuse in Czech illegal puppy mills. In the absence of data or research evidence, I was relegated to information provided by lawyers, judges, field workers, charities and also journalists. The third section is devoted to an outline of existing Czech legislation and the (inadequate) ways it sanctions animal abuse and illegal puppy mills. It also presents issues of practical implementation of these regulations. The central part of the study analyses the newly adopted legislation, namely an amendment to the Criminal Code tabled in the Chamber of Deputies in mid-2018 which, following a March 2020 discussion of a Senate members’ amendment by the Chamber, was signed into law by the Czech President. It basically provides for tougher sentences against abuse of animals through inadequate breeding conditions and defines so-called deficient puppy mills. This is a legal breakthrough given the magnitude of the problem of dog and cat breeding establishments in the Czech Republic. In its analytical section, my study focuses on the debates carried out in both chambers of the Czech parliament. Those debates reflected the fundamental arguments in favour of and against the new regulation. The analysis will concentrate on the following:(1)Pivotal arguments by criminal law experts that sparked intensive public debate and were subsequently deployed in the parliamentary debates.(2)Arguments in favour of and against tougher penalties for animal abuse and the so-called deficient puppy mills.

### 3.3. The Data Analysed and the Criminal Code Amendment

#### 3.3.1. The Data

As stated in the Introduction, the Czech Republic is a country of thriving puppy mill business. It has repeatedly failed to toughen its laws against cruel abuse and inhumane treatment of animals in commercial breeding establishments. Charities have succeeded in highlighting the problem in social discourse, yet the issue was long ignored in politics. The following analysis of parliamentary debates will help in outlining the competing coalitions that emerged in the process of debating the Criminal Code amendment in 2018–2020 [39] and the different arguments they employed. The debate process spanned a relatively long time period: it started on 22 June 2018, and it was not until 12 February 2019 that the first reading of Document for Discussion No. 214/0 [43] began in the Chamber of Deputies (see verbatim transcriptions) (see Table 1). The Chamber concluded its debate on 12 December 2019. The Senate then debated the bill, as Document for Discussion No. 181/0 [44], at its plenary on 30 January 2020 and reversed the Chamber’s position. Finally, the Chamber agreed on the Senate amendment and passed tougher sentences at its plenary on 3 March 2020. The resulting provisions were nothing new because the Senate had merely embraced the original members’ bill of 77 Deputies that had been changed by the Chamber in the course of its debate. The group of 77 draftsmen included comparable numbers of coalition and opposition MPs (with the opposition accounting for 47 of its members).

Tougher criminal penalties are going to be applicable to Maltreatment of Animals and operation of so-called puppy mills. Maltreatment perpetrators will face up to six years of imprisonment instead of five. In extreme cases of breeding animals in inadequate conditions jeopardising their life, the courts will be able to impose prison sentences of up to ten years. The version originally passed by the Chamber of Deputies was reduced to allowing courts to impose a separate punishment of Prohibition to Rear Animals (for up to 10 years for individuals and up to 20 years for businesses). One reason it was opposed to toughening the sentencing guidelines was that jurists argued the prison terms were too long compared to those for violence against human victims, an argument that strongly resonated during the debates in both parliamentary chambers.

The analysis focuses primarily on the positions of criminal law experts that influenced the subsequent debates in both parliamentary chambers. There were two main pieces of legal analysis: one by President of the Supreme Court Pavel Šámal that was opposed to interventions in the Criminal Code in general and tougher sentences in particular; and the other by attorney Robert Plicka, a long-term animal rights and protection specialist. These positions played central roles in the subsequent debates on such toughening of the sentencing guidelines for animal abuse that was unprecedented in the Czech system of criminal law.

Parliamentary debates provide the second source of data for my analysis. The opposition was highly active in debating this bill (the Czech Republic is governed by a minority government of PM Andrej Babiš’s ANO (Action of Dissatisfied Citizens) and the Social Democracy which has been tolerated by the Communist Party; six other parties are in the opposition). In the course of the different Chamber of Deputies readings, floor speeches were given by 13 coalition and 25 opposition MPs. At the same time, the original members’ bill to amend the Criminal Code was supported almost equally by opposition and collation Deputies. There was no coalition–opposition cleavage on this issue and, instead, the line was drawn between different attitudes to animal abuse. The debate in both parliamentary chambers was accompanied by activist lobbying, with charities sending to MPs emotional testimonies of cruel abuse and other materials. My reconstruction of the key arguments for and against the proposed toughening of animal abuse provisions is based on verbatim transcriptions from both chambers.

#### 3.3.2. The Criminal Code Amendment

Precisely in view of the above-mentioned, the illegal activity of operating puppy mills is difficult to detect and prosecute. The same applies to the entire post-communist region, which is much more opposed to toughening these norms than Western Europe. However, in June 2018, a members’ bill to amend the Criminal Code was drafted by as many as 77 MPs (of the total of 200) from both coalition and opposition parties. Its goal was to make the protection of animals against cruelty more effective, to toughen the sentencing guidelines for Maltreatment of Animals, and to provide for adequate criminalisation of puppy mills. This first version foresaw a new punishment of “Prohibition to Possess and Rear Animals” (applicable to legal persons as well), including penalties for obstruction of justice. The longer prison terms were central: those for Maltreatment of Animals were increased by a margin of one to three years (to three, five and eight years, respectively, depending on qualification of the offence). A new separate criminal offence was defined for deficient puppy mills, the Breeding of Animals in Inadequate Conditions, with prison sentences for the different qualifications at 1, 4 and 8 years for felonies and 10 years for especially serious felonies, as well as criminal liability for Non-Prevention of Criminal Offence.

The Government took a neutral position on the bill, arguing that it failed to address the handling of animals in case of imposing the punishment of Prohibition to Possess and Rear, that insufficient provisions were made for compliance checks, and that some other implications had not been taken into consideration. The Chamber of Deputies Committee on Constitutional and Legal Affairs undertook a complex revision of the bill and tabled a new wording in the Chamber that addressed the Government’s comments (providing for enforcement proceedings for the new punishment of Prohibition to Possess and Rear Animals, entrusting compliance checks to regional offices of the Veterinary Administration instead of the Probation and Mediation Service, allowing the courts to impose this sentence by issuing a Penal Order or to invoke a Conditional Waiver of Execution of a Remaining Portion of a Sentence, Limitation of Criminal Liability, etc.) It deleted the separate criminal offence for deficient puppy mills and instead inserted it as a second basic definition under the existing offence of Maltreatment of Animals. It fundamentally refused any toughening of imprisonment sentences. This gave rise to a future conflict among criminal law experts that will be mentioned below. The Committee’s central argument was the need to maintain a balanced system of punishments in the existing Criminal Code and prevent its destabilisation through one-sided toughening of the sentencing guidelines.

In the second reading, some of the former draftsmen (a group of 15 MPs) tabled the so-called compromise amendment to correct the revision proposed by the Committee on Constitutional and Legal Affairs. The amendment again included almost all of the toughened sentencing guidelines for Maltreatment, namely three, five and six years of imprisonment depending on qualification of the offence. At the same time, the threshold of criminal liability was lowered by defining the basic offence of Maltreatment as merely “cruel or agonising” instead of the former “especially cruel or agonising”. Furthermore, the compromise amendment brought back the separate criminal offence for deficient puppy mills, with their former sentencing guidelines, and included a definition of the gravest forms of this offence (Paragraph 4). The least grave form (Paragraph 1) referred to deficient puppy mills without the trait of material profit (business).

The fierce competition between both amendments (that by the Committee on Constitutional and Legal Affairs and the compromise one) led to an “agreement” between their proponents (concessions by both parties were presented). However, in the third reading, the Chamber of Deputies lent majority support to the Committee’s amendment as a whole. In the key voting session No. 382, 80 members abstained, and the compromise amendment fell six votes in favour short of passing. At the 39th Plenary Meeting of the Chamber of Deputies, the final voting session No. 382 on the Bill of 18 December 2019 was attended by 186 members, of whom 176 were in favour and none were opposed.

An interesting twist occurred in the Senate. Although the tougher sentencing guidelines had been voted down by the Chamber, the Senate recommended increasing the prison terms up to six years for Maltreatment and up to 10 years for international puppy mills. Thus, it effectively reintroduced the Deputies’ former compromise amendment. The bill, along with the Senate’s amendment, travelled back to the Chamber. It eventually obtained support from PM Andrej Babiš, which was crucial for securing the votes of the Chamber’s strongest caucus of ANO. As a result, the second attempt was successful, and the toughening of sentences passed in the Chamber of Deputies. Another round of rather heated debate ensued. There were once again two clear competing positions on providing a legal solution to the problem of ineffective protection of animals from abuse, whereas the more stringent norm prevailed. The introduction of tougher sentencing guidelines for these offences will allow the police to effectively detect illegal puppy mill networks by deploying wiretaps and other so-called operative investigative means. Criminal proceedings will not result in so-called diversions, i.e., alternative sentences for which only petty sanctions can be imposed. From now on, perpetrators of these offences will face the risk of unsuspended prison sentences.

Several fundamental changes have resulted from the tougher sentencing guidelines:(1)In the most brutal cases, judges will be free to consider imposing unsuspended prison sentences (see restriction prescribed in Section 55 (2) of the Criminal Code).(2)Conditional Stay of Criminal Prosecution will not be possible.(3)The regime of so-called Penal Orders foreseen for misdemeanours, under which a maximum suspended prison sentence of one year can be imposed and the perpetrator does not even have to present himself to the judge, will not be applicable.(4)Offenders will no longer have access to Waiver of Punishment.(5)In cases of the most brutal and organised forms of Maltreatment of Animals (i.e., illegal puppy mills with profits in the order of millions), wiretaps will be deployable to dismantle hidden organised networks of breeders.(6)House search warrants will be easier to issue (since illegal breeders keep their animals in dwellings which cannot be entered by the police without a house search warrant and courts are reluctant to issue such warrants for mere misdemeanours).(7)The worst cases will not be decided by single judges but by better-qualified court chambers.(8)The importance of the value of protecting animals against intentional abuse will be communicated,(9)Better protection of animals against intentional abuse will also improve the protection of people, because there is statistical evidence that individuals who abuse animals at a young age will often turn their violence against human victims later in life [1,45].

#### 3.3.3. Criminal Law Experts’ Arguments

Two competing streams of opinion emerged among experts during the parliamentary debate on toughening the criminal sanctions. The first stream consisted of proponents of harsher sentences led by attorney Robert Plicka, the author of the proposed Criminal Code amendment to increase the prison terms for Maltreatment of Animals and criminalise illegal puppy mills. Note that their version eventually passed. Their opinion was strongly opposed by a stream led by Deputy Marek Benda and Senator Miroslav Antl, both chairmen of their respective chambers’ Committees on Constitutional and Legal Affairs. These leaders were against tougher guidelines because, as they argued, penalties for cruel treatment of animals should not be harsher than or comparable to those for cruel treatment of people. They pleaded for judicious introduction of amendments to the Criminal Code in order to maintain the ultima ratio principle. This stream of opinion was joined by the arguments of new Constitutional Justice Pavel Šámal, a criminal law expert and a long-term President of the Supreme Court. When the bill was being debated by the Senate, Justice Šámal sent to all members of parliament his legal analysis of the bill and warned against toughening sentences for animal abuse.

I am going to highlight four of those criminal law experts’ arguments. The same arguments also occurred, at different levels of intensity, during debates in other countries of the post-communist region.

## 4. Results

### 4.1. Argument 1: Animals Will Be Protected More than Humans

This was a frequently used argument against the tougher guidelines. To quote Justice Šámal: ‘The existing criminal legal provisions on protection of animals against cruelty, even after being complemented in accordance with the amendment passed by the Chamber of Deputies, are basically in line with the ultima ratio principle and the fact that in the field of animal protection, criminal law and its repressive means should always be the last resort, the uttermost means of legal protection’ [46]. Justice Šámal goes on to criticise the fact that the proposed provisions ‘bestow such privileged protection upon animals that it is, in some respects, even stronger than the protection of human life and health.’ To support his case, he stated (and his argument was later repeated by speakers in the Chamber debate) that the most moderate case of Harm to Health out of Excusable Motives can be punished with up to one year’s prison, compared to the proposed maximum sentence of up to three years of imprisonment for the most moderate case of Maltreatment of Animals. Justice Šámal also decries the absence of in-depth criminological research and of an assessment and comparison with other types of crime (the latter due to the fact that the members’ bill had not undergone the standard legislative procedure and had not received comments from relevant government authorities).

The general argument of the opponents of tougher sentences was that unsuspended prison sentences that are foreseen in the proposed Criminal Code amendment should only constitute the ultimate response to animal abuse; that the current trend was, quite contrarily, to educate and reform perpetrators [47]. However, the main proponent of tougher sentences, attorney Robert Plicka, who undertook extensive lobbying among lawmakers when the amendment was on their agenda, argues that ‘enhanced protection of animals neither damages nor weakens the level of protection bestowed upon humans. In contrast, Professor Šámal views the issue in purely legal terms (and without regard for the empirical context and the level of pathological behaviour among these individuals), and above all, he is not familiar with sociological studies that clearly demonstrate that animal abuse is a precursor to abuse of human victims. Conversely, increased animal protection simultaneously enhances the protection of people.’ [41]. He argues that Justice Šámal mistakenly compares acts that are different and concludes that animals will be protected in a similar way as people. ‘For instance, Professor Šámal compares negligent homicide to intentional maltreatment of animals resulting in death as an example of imbalance. However, negligent killing of an animal is an administrative, not criminal offence. This is therefore a biased comparison of different acts. Similarly, there is a substantive difference between harming a person out of Excusable Motives (for example, due to their Previously Condemnable Conduct) and intentionally (that is, purposely) Maltreating Animals in an Especially Cruel or Agonising Manner (!). This is not the same act, and therefore, such a comparison is again a biased one.’ [41].

There are two competing perspectives here: a conservative and strictly juristic one represented by the Supreme Court President, and an activist and conservative one that takes into consideration not only law but also sociological aspects, as well as the fact that Czech authorities have been unable to effectively enforce laws against animal abuse, including illegal and inhumane puppy mills. A fact check with the Criminal Code clearly refutes the argument that penalties for cruel treatment of animals are more stringent than with human victims. Basic definitions of offences cannot be compared to so-called qualified definitions (those associated with tougher penalties than the basic definition), intentional offences to negligent ones, offences committed through action to ones committed through inaction, etc.

### 4.2. Argument 2: Penalties for Animal Abuse Will Be More Stringent than in the Case of Murder of a Newborn Child by Its Mother

This strongly resonated among the opponents of tougher sentences during the parliamentary debate. Once again, though, comparable things should be compared. Three types of offences are defined in the Criminal Code: (i) basic, (ii) qualified, and (iii) privileged. Murder of a New-born Child by its Mother is not a basic offence but a privileged one, taking into account the mother’s “state of mental disturbance immediately after the child’s birth”, and this is why the penalty is relatively soft (three to eight years of imprisonment). Under this privileged offence, the underlying basic offence of Murder (10–18) years is modified by a fact that makes the offence less grave. In contrast, the basic offence of intentional Maltreatment of Animals in the amendment at hand is punished with six months to three years of imprisonment. It is only in the very last paragraph, for the gravest qualified definition of this offence, that a sentence of two to eight years of imprisonment is foreseen (which is still lower than the above-mentioned privileged offence of murdering one’s new-born). It is necessary to compare basic definitions with basic ones, privileged ones with privileged ones, etc.

### 4.3. Argument 3: By Passing the Amendment, the State Would Violate the Ultima Ratio Principle. It Should Opt for Other Means of Protection, Including Administrative Law

President of the Supreme Court Pavel Šámal argued: ‘The means of administrative law have much broader uses here, and they have to be utilised because I do not find it suitable to rely exclusively on criminal prosecution and making it more stringent’. Contrarily, the proponents of toughening led by Robert Plicka argued that the ultima ratio principle was enshrined in Section 12 of the Criminal Code. This principle of penal law prescribes that criminal penalties should only be applied when all other methods of liability (civil law, administrative law etc.) fail. The amendment would by no means violate that principle because Section 12 would remain intact. Existing means of administrative law are absolutely inadequate and impotent. The actual fines imposed for animal abuse range in hundreds or thousands of euros at most, even though the maximum amount is CZK 500,000 (EUR 18,000). Such small pecuniary penalties cannot be expected to disrupt economic operations in the order of hundreds of thousands of euros. Therefore, given the magnitude of the social harm caused by current forms of crime against animals, criminal law is the only effective way to prevent them.

Although the prison terms for animal abuse in neighbouring countries (Slovakia, Austria, Germany) are not as long, such a comparison is complicated. The former was noted by Justice Šámal (2020) in his analysis of the legislative bill. The counterargument was that a comparison of prescribed prison terms in the Czech Republic and abroad is not quite relevant because each system of criminal law has its specifics and in the case of less serious criminal offences, the more important question is whether and to what extent the country’s laws allow for imposing unsuspended prison sentences at all.

As outlined above, the Czech law prescribes that first-time offenders can almost never obtain prison sentences for offences where the upper limit of the prescribed prison term does not exceed five years, even if they do not regret having committed that criminal offence. The implication is that offences with much shorter prison terms for which one can actually be sent to prison are effectively more severe than offences with longer terms in cases when unsuspended prison sentences cannot be imposed. Some people say that a nation’s level of development can be seen in the way it treats animals. In addition, where else should cruelty to animals and puppy mill activities be eliminated aggressively than at their source, namely in the Czech Republic, where the illegal supply of puppies to all of Europe originates? Other countries of the world, too, have opted for tougher sentencing guidelines for animal abuse, both in and outside Europe.

### 4.4. Argument 4: Increased Maximum Prison Terms Will Undermine the Existing System of Sentencing Guidelines While They Do Not Automatically Have Preventive Effects

The Special Part of the Criminal Code, which defines individual offences and sentencing guidelines, consists of 13 chapters ordered by the importance of the different social values protected by criminal law. Even after the amendment, the offence of Maltreatment of Animals will remain in the same chapter (Criminal Offences against Environment) and within that chapter, its maximum prison terms will not substantially deviate from other offences defined therein [48]. In addition, the proponents of tougher guidelines argue that the way the offences under a single Criminal Code Chapter are organised cannot prevent the legislators from bestowing stronger protection on a public interest of their choice. In this case, longer prison terms result primarily in the ability to deploy Operative Investigative Means, impose unsuspended prison sentences, etc., which will undoubtedly enhance the preventive effect. Then the counterargument of the critics led by Justice Šámal is that ‘an unsuspended prison sentence should only be imposed as a truly last resort, with discretion in responding to different cases of maltreatment of animals’ [46].

### 4.5. Arguments in Favour of and against Tougher Penalties for Animal Abuse and Deficient Puppy Mills

Above all, none of the speakers in either parliamentary chamber diminished the gravity of animal abuse. While all agreed that these criminal offences should be prosecuted effectively, they differed on the ways to achieve tougher regulation.

#### 4.5.1. Arguments by Those in Favour of Tougher Penalties

It was repeatedly claimed during the debate in both chambers that the Czech Republic is shamefully referred to as “Europe’s puppy mill”. This was mentioned by opposition and coalition MPs alike. Administrative measures are ineffective and so are the existing measures of criminal law. Statistics indicate that the most brutal instances of abuse and the suffering of hundreds of animals in puppy mills have resulted in suspended sentences at best, which, according to proponents of tougher regulation, indicates clearly undervalued penalties. Criminal proceedings are hampered by waiting for expert opinions to clarify whether an animal really suffered in the gravest manner (criminal offence) or “merely” in an agonising manner (administrative offence).

The position of Deputy Markéta Pekarová Adamová perhaps best represents the group of draftsmen: ‘Finally, we intend to criminalise these practices, which are absolutely unacceptable, and to provide law enforcement with better means to detecting them. Note that this criminal offence will in no way affect negligent culpability or omission on the part of fair breeders or farmers because this is an intentional criminal offence threatening animals’ lives’ [49]. Rapporteur Jan Žaloudík advocated for the tougher guidelines in the name of the Senate: ‘It is not only the mental health of individuals that is at stake but also the mental health of society; deviants must be fundamentally opposed. To those saying that we don’t understand criminal law, that we’re somehow changing the proportions, I say yes, we are. We don’t. The things we speak about on the Senate floor are not always things we understand, just like the three-hour debate was not one among virologists. And the same applies to the Senate. But we believe that this issue is so clear in terms of public interest that I was grateful to the 77 Deputies who put this bill on the table in June 2018. And things have been debated responsibly for a year and a half and thus consensus has been reached among you both in the Agriculture Committee and in the Committee on Constitutional and Legal Affairs. Although the voting then did not go well when 87 Deputies were in favour, 17 against, that is in favour of the new version, and 80 abstained. So we thought that this is an opportunity to help the other part of the Parliament, the Chamber of Deputies, and submit the same amendment—and this is the same wording that in fact originated from you. In our institution, this was the initiative of Senator Hraba and 21 other senators. So it should be submitted once again for consideration here in the honourable Chamber of Deputies so that things take a new turn.’ [50].

Provided living animals are acknowledged as not being things, it is reasonable to provide them with protection beyond that bestowed upon assets. The use of effective instruments is curbed for offences with short prison terms (misdemeanours): courts practically cannot impose unsuspended prison sentences on first-time offenders, are motivated to apply Penal Orders (the perpetrator does not even stand before a judge and the maximum term of unsuspended sentence is one year), the seeming pettiness of the offence may even prompt Conditional Stay of Criminal Prosecution, misdemeanours leave judges unwilling to order house searchers (key to gathering evidence of maltreatment), and wiretaps cannot be deployed against organised and hidden activities of illegal puppy mills because this is only allowed for prison terms of eight years or longer.

#### 4.5.2. Arguments by Opponents of the Criminal Code Amendment

The ultima ratio principle informs us that criminal prosecution should not be a means to solving everything. Opponents of tougher regulations, led by chairmen of the respective Chamber and Senate Committees on Constitutional and Legal Affairs, found it necessary to create an environment that eliminates animal abuse by developing more effective means on the part of government administration (regulation, prevention, inspection systems) (see Table 2).

In this argument, animals should be protected, not made human. Accordingly, different sentencing guidelines should be applied to protection of the life and health of humans and animals. However, prison terms in excess of five years are not prescribed for any of the basic offences categorised as Endangering Life or Health (of humans), namely Cruel Treatment, Failure to Provide Assistance, Spreading of Contagious Disease, etc., or for Maltreatment of Entrusted Person.

Member of the oppositional Civic Democratic Party and former Minister of Agriculture, Petr Bendl, supported this argument in the Chamber of Deputies: ‘What are the implications of this bill? In my view, it rather demonstrates hopelessness because the government administration, which has sufficient legislative means to handling this, has been rather inactive. Indeed, someone rearing a large amount of animals—rearing is perhaps a euphemism here—in puppy mills, is typically undertaking an unauthorised business activity and is typically not paying any taxes. And this is where I think the government administration has instruments to enforce those things and to be active. Unfortunately, the public has not seen any such activity, and that’s a pity. Tax authorities, I think, and perhaps even the Ministry of Finance should realise that this might be the way for us to handle puppy mills.’ [51]. His speech was complemented by a member of the same Chamber caucus and Chairman of the Committee on Constitutional and Legal Affairs, Marek Benda: ‘Let me just say that the Senate had a relatively extensive analysis by Supreme Court justices led by Professor Šámal, a recently elected member of the Constitutional Court, telling them: Don’t do this. This is not a way to make legislation, you are destabilising, blurring the boundaries of meaningfulness in the Criminal Code not only in terms of human life, human health and other things but also in terms of Offences against Environment and protection of animals and others’ [52].

According to these opinions, substantive criminal law, as opposed to other bodies of law, is constructed as a highly interrelated and balanced system. Critics argue that a deviation from (growth of) prison terms may destabilise the system. Metaphorically, one cannot see the wood for the trees. The deviation would motivate the undesirable toughening of sentencing guidelines for other categories of criminal offences (by type of protected interest). In other words, the opponents refused the concept of exemplary sentencing. They clearly welcomed the introduction of a new specialised punishment of Prohibition to Possess and Rear Animals. They also noted that when first-time offenders exhibit negative conduct, their suspended or alternative sentences will be transformed into unsuspended prison sentences, or they will commit recidivism; thus, in the right cases, criminal law will achieve the “end of the road” effect eventually. They were also concerned that a separate criminal offence for deficient puppy mills might be abused by “activists” for large-scale attacks against intensive animal farming etc. This concern was subsequently viewed as ill-grounded because the definition of the criminal offence was worded in a sufficiently unambiguous way.

## 5. Discussion

The Criminal Code amendment on animal abuse was debated in a relatively passionate spirit. First, there were two competing legal positions. That opposed to changes argued that tougher sentencing guidelines for Maltreatment of Animals would destabilise the existing system of punishments, and any toughening should cover a wider array of offences, including those involving abuse of human victims. ‘While I love animals, I place human life above them.’ [53]. Promoted especially by MPs with legal education, this line of argument completely ignored the evident fact that animals were being abused and none of the perpetrators had faced an unsuspended sentence to date.

This was opposed by the other group that did not always use strict and elaborate legal arguments and instead embraced the arguments of the civic sector. Members of this group promoted the idea that it is also in people’s own interest to toughen the sentencing guidelines for animal abuse, that not doing so would ‘undermine our systems of values’ [54]. They often employed metaphors such as: ‘You would be reluctant to entrust your own child’s babysitting to a deviant who has treated an animal inappropriately’ [54]. They argued that since the system of penalties for animal abuse was not working, it was necessary to toughen the sentencing guidelines for Especially Cruel or Agonising Maltreatment of Animals resulting in permanent damage to the animal’s health or its death, and to make it possible for courts to impose the punishment of Prohibition to Possess and Rear as well as unsuspended prison sentences. Some also viewed the systematic toughening of sentencing guidelines preventively as a means to deterring perpetrators [55].

The second focus of the animal abuse debates was on issues of public health and mental health in society. Some MPs found the everyday fact of trafficking in animals and cruel, inhumane breeding conditions in the Czech Republic so problematic that it “affects the mental and social health of our society” [56]. The opponents generally admitted that animal abuse should be addressed as a pathology but argued that instead of toughening the sentences and effectively destabilising the Criminal Code, one should enhance the application of existing instruments of administrative law and penalties by the Veterinary Administration. However, those instruments had proven ineffective in the field and left the authorities unable to furnish enough evidence to prove the criminal offences.

All in all, Czech lawmakers did view animal abuse as a problem and were able to condemn it unequivocally, as something that does not belong to the Czech society. Yet they were able to spend hours quarrying about the ways to enforce laws against such crime. One group supported an ad hoc toughening of sentencing guidelines, which has become law in the Czech Republic, while the other warned against it and preferred a complex change of the Criminal Code to prevent undermining the hierarchy of punishments with respect to human victims.

## 6. Conclusions

RQ1: What are the main legislative issues of and obstacles to providing a legal solution to the problem of illegal puppy mills in the context of post-communist countries?

This paper is conceived as a case study of a recently debated change in Czech legislation. Similar debates have been held in other countries of the post-communist region as well. Discussions by experts, above all, but also by politicians in EU institutions, may help gradually transfer to legal systems of post-communist European countries such instruments that might eliminate animal abuse and illegal puppy mills more effectively [57]. The goal is also to make the instruments enshrined in these legal provisions effective and truly helpful in addressing this illegal activity. Three main obstacles have been identified in the analysis of the positions of criminal law experts and lawmakers’ speeches during the debate in both parliamentary chambers. First, there is reluctance to admit the gravity of the problem of cruel animal abuse and deficient puppy mills. For several years now, Czech politics has been confronted with warnings by the civic sector that trafficking in animals is a considerable problem and breeders are taking advantage of under-regulation. This is evidenced by the fact that the Criminal Code amendment bill was tabled by a group of MPs, rather than the Government and the ministry responsible. As a second problem, important political leaders who are able to change sentencing guidelines have been motivated by legal conservatism to take a restrained approach, even if they realise the problem’s urgency. Related is the third problem, namely that the Criminal Code amendment leads to inconsistencies, and in particular violates the ultima ratio principle, as noted by a number of criminal law experts. The latter also warned that the changes enacted would not work in practice.

RQ2: How did criminal law experts opine about toughening the penalties against animal abuse and deficient puppy mills, what kind of arguments were formulated, and how did they shape the subsequent decision making by lawmakers?

There are proponents as well as opponents of the changes. Proponents argue that change is demanded by the people and required to protect both public health and a healthy society. They deem it necessary to introduce a new criminal offence of breeding in inadequate conditions, i.e., puppy mills, so that law enforcement can detect such establishments. Since the situation was underestimated, consistent prosecution can now only be made possible by the threat of unsuspended prison sentences.

Contrarily, tougher sentencing guidelines for animal abuse cannot be introduced in this way for two reasons: first, criminal penalties should remain the last resort of legal protection and, second, one must prevent a disproportionately privileged protection of animals. Opponents of tougher guidelines argue that the new legal provisions place these criminal offences on par with the protection of human life and health.

RQ3: What were the arguments of Czech politicians in favour of or against toughening criminal penalties against animal abuse?

The wording of the bill that arrived in the Senate failed to reflect any of its original goals—not one sentencing guideline would be toughened. Yet, it was precisely the shortness of existing prison terms that made wiretaps and house searches impossible and, thus, prevented law enforcement from proving even a single case of animal abuse, including puppy mills. These highly organised activities involve many individuals, including veterinarians, couriers, breeders, clubs, etc. The opponents of tougher sentencing guidelines sided with the criminal law experts whose arguments were described in the previous paragraph. The proponents referred not only to the above-mentioned but also to the arguments of some judges that they were unable to punish more effectively by imposing unsuspended prison sentences, to the importance of protecting animals, and to public health considerations. Finally, there was one undeniable argument, namely, that while animal trafficking is the third most lucrative illicit market in Europe, it involves cruel abuse of animals in their reproduction, smuggling, etc., and sadly, the Czech Republic is playing a highly conspicuous role in this business.

## Figures and Tables

**Table 1 animals-10-01020-t001:** The process of debating tougher criminal penalties in both chambers of the Czech parliament.

Debate	Reading	Date	Floor Speakers (Coalition/Opposition)
**Chamber of Deputies**	1st reading	12 February 2019	4/11
	2nd reading	26–27 November 2019	5/9
	3rd reading	18 December 2019	4/5
**Senate**	Single reading	30 January 2020	10/22

Source: The Chamber of Deputies and the Senate, own elaboration.

**Table 2 animals-10-01020-t002:** Arguments in favour of and against tougher penalties for animal abuse and deficient puppy mills.

In Favour	Against
(1) The Czech Republic is referred to as “Europe’s puppy mill”	(1) Criminal prosecution should not be a means to solving everything.
(2) Administrative measures are ineffective and so are the existing measures of criminal law	(2) Animals should be protected, not made human. Accordingly, different sentencing guidelines should be applied to protection of the life and health of humans and animals.
(3) Criminal proceedings are hampered by waiting for expert opinions to clarify whether an animal really suffered in the gravest manner (criminal offence) or “merely” in an agonising manner (administrative offence)	

Source: own elaboration.

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
