# Peer review of "How to Fight Puppy Mills: Toughening the Sentences for Animal Abuse in the Post-Communist Region"

_animals, 2020, doi:10.3390/ani10061020_

Round 1
Reviewer 1 Report
REVIEW: How to fight puppy mills? Toughening the sentences for cruel animal abuse in the post-communist region
I found this paper to be very interesting – I learned a lot from reading it which is definitely in its favor. I think there are several revisions necessary for the paper to have maximum impact particularly to audiences across disciplines. My comments below are in order of appearance in the paper:
- The title focuses on the post-communist region. There are a few mentions of situations in other post-communist countries but there is not enough such discussion to really say that the paper focuses on the region or is applicable outside of the Czech case. One way to deal with this is to change the title but I would very much prefer to have the authors add discussion on what is going on in other post-communist countries or how the post-communist case makes the application of puppy mill law unique. Along the same lines it would be good to see discussion of how the law discussed here compares to other international cases of puppy mill regulation. If it would be possible a table or appendix comparing some sample laws would be an excellent addition.
- As an animal welfare researcher and advocate I want to raise the next issue with caution. It is clear that puppy mills are reprehensible. However, I think for an academic journal article some of the value language needs to be softened a bit – I’m thinking of terms like “catastrophic,” “reprehensible”, “alarming,” “must be condemned by society.” Again, I completely agree but if slightly different language could be used that would be good.
- On a small editing note, the authors refer to the sections of the paper as “chapters” – this should be changed to sections.
- On page 4, lines 161-174: are there literature citations that can be added in this paragraph?
- My most important comment is that the description of the analysis of the debates is long and complex and somewhat difficult to read and process. I expect that readers not in the legal field might have particular problems with it. It would be extremely helpful if the authors could add a table or figure at the end of this discussion which could summarize something like: the values/concepts discussed, positions taken on each, the legal basis for each, who supported them, and then what made it in the final law.
Completely unrelated to any changes in the paper: it got me to thinking about the potential transmission of diseases from exported puppies to animals in the importing country and then on to humans. The authors also noted that puppy mill exporters tended to be involved in other crime – I wonder if the same people are involved in illegal exports of both drugs and puppies. Just curious – nothing needs to be added but as I said at the beginning, the paper was very interesting.
Author Response
Reviewer 1
Point 1:
The title focuses on the post-communist region. There are a few mentions of situations in other post-communist countries but there is not enough such discussion to really say that the paper focuses on the region or is applicable outside of the Czech case. One way to deal with this is to change the title but I would very much prefer to have the authors add discussion on what is going on in other post-communist countries or how the post-communist case makes the application of puppy mill law unique. Along the same lines it would be good to see discussion of how the law discussed here compares to other international cases of puppy mill regulation. If it would be possible a table or appendix comparing some sample laws would be an excellent addition.
Response 1:
One whole paragraph on this topic has been added (lines 162-167).
Point 2:
As an animal welfare researcher and advocate I want to raise the next issue with caution. It is clear that puppy mills are reprehensible. However, I think for an academic journal article some of the value language needs to be softened a bit – I’m thinking of terms like “catastrophic,” “reprehensible”, “alarming,” “must be condemned by society.” Again, I completely agree but if slightly different language could be used that would be good.
Response 2:
I have reduced or removed these expressive terms. Thank you for this comment!
Point 3:
On a small editing note, the authors refer to the sections of the paper as “chapters” – this should be changed to sections.
Response 3:
Changed to “section”
Point 4:
On page 4, lines 161-174: are there literature citations that can be added in this paragraph?
Response 4:
See FN 20 and 21, please.
Point 5:
My most important comment is that the description of the analysis of the debates is long and complex and somewhat difficult to read and process. I expect that readers not in the legal field might have particular problems with it. It would be extremely helpful if the authors could add a table or figure at the end of this discussion which could summarize something like: the values/concepts discussed, positions taken on each, the legal basis for each, who supported them, and then what made it in the final law.
Response 5:
I inserted Table 2. Arguments in Favor of and Against Tougher Penalties for Animal Abuse and Deficient Puppy Mills.

Reviewer 2 Report
A timely and interesting article about animal abuse and puppy mills in a location that academic and non-governmental research has shown to be a problem. There are areas for improvement. Legally, animals are usually considered as property, so the mention of moving assets near the beginning needs to be explained earlier.
Line 132-35 "Meanwhile, countries of the former communist bloc have seen extensive debates on whether and how to criminalise. This, along with the state of debates on women’s rights or minority rights, reflects a strong heritage of 40 years of communism." The first sentence needs to be cited and expanded upon and it is unclear how the second sentence is relevant.
Line 137-138 needs to be cited and nuanced. There is little proof that puppy trafficking and breeding is organised crime in the sense of gangs or mafias. The trade is structured, but no evidence indicates involvement of organised crime groups.
Line 159 - the source you cite says nothing about your claim that Europe's third largest black market is puppies.
Page 6 - it is a huge unsubstantiated assumption to say puppy mills continue because of the sanctions. There could be lots of additional explanations and motivations.
In general, the grammar and word choice need to be more thoroughly proofread. It is unclear why 'we' is being used throughout if this is a single authored paper. As this is an article (not a book), it doesn't have chapters. There are sections.
Overall the article is assertive rather than evidenced. Much more literature needs to underpin claims made particularly about the nature of puppy mills and post communist societies. There are a dozen reports from charities in Europe about puppy mills that should be cited (Dogs Trust, Eurogroup for Animals etc.).
The main area for improvement is the 'so what' question. Whilst the information is interesting, what does it help with in society? Knowing that this is how law is formulated what does that teach about society and why is that important?
Author Response
Reviewer 2
Point 1
A timely and interesting article about animal abuse and puppy mills in a location that academic and non-governmental research has shown to be a problem. There are areas for improvement. Legally, animals are usually considered as property, so the mention of moving assets near the beginning needs to be explained earlier.
Response 1:
I have added the lines 123-127.
Point 2:
Line 132-35 "Meanwhile, countries of the former communist bloc have seen extensive debates on whether and how to criminalise. This, along with the state of debates on women’s rights or minority rights, reflects a strong heritage of 40 years of communism." The first sentence needs to be cited and expanded upon and it is unclear how the second sentence is relevant.
Response 2:
Both sentences have been removed.
Point 3:
Line 137-138 needs to be cited and nuanced. There is little proof that puppy trafficking and breeding is organised crime in the sense of gangs or mafias. The trade is structured, but no evidence indicates involvement of organised crime groups.
Response 3:
I understand this criticism. I have adjusted the statement - again with a view to removing expressiveness.
Point 4:
Line 159 - the source you cite says nothing about your claim that Europe's third largest black market is puppies.
Response 4:
I changed the source (see Ref. 19).
Point 5:
Page 6 - it is a huge unsubstantiated assumption to say puppy mills continue because of the sanctions. There could be lots of additional explanations and motivations.
Response 5:
I understand. This applies to the paragraph (“Cruelty to animals and operation of deficient puppy…“). I have modified the argument so that this does not follow.
Point 6:
In general, the grammar and word choice need to be more thoroughly proofread. It is unclear why 'we' is being used throughout if this is a single authored paper. As this is an article (not a book), it doesn't have chapters. There are sections.
Response 6:
I removed “we” (typical for texts in Czech language – sorry) and changed “chapters” to “sections”.
Point 7:
Overall the article is assertive rather than evidenced. Much more literature needs to underpin claims made particularly about the nature of puppy mills and post communist societies. There are a dozen reports from charities in Europe about puppy mills that should be cited (Dogs Trust, Eurogroup for Animals etc.).
Response 7:
I added the positions of non-profit organizations (eg in the list of literature 15), The German Vier Pfoten foundation, One Green Planet and The Humane Society of the United States (see Ref. 16, 19-21).

Reviewer 3 Report
This is a very important and up until now often neglected area of animal welfare policymaking, so I would like to encourage the author to proceed with the research agenda. The study provides a detailed analysis of the policy process around Czech legislation and the different views of actors on this process. However, I think the study as it is, lacks a theoretical embedment and the connection to the according scientific debates.
In particular, I would like to raise the following points:
What are the possible conclusions or lessons for other regions from this single case study?
Did the author consider the initiative of the EU on illegal dog and cat breeding? How do these come into play? (https://www.europarl.europa.eu/news/en/press-room/20200206IPR72016/stop-illegal-trade-in-cats-and-dogs-says-european-parliament)
The manuscript includes several normative assumptions, these should be eliminated or at least be supported by some literature, e.g. page 1 lines 41/42, but also in the remaining part of the manuscript. In addition, at some points there is a rather colloquial wording (e.g. p. 2 lines 49), which should be revised. Also, I would recommend proof-reading by a native speaker.
What is the theoretical and disciplinary background? On page 2 it is states that “a legal and political” one, is this political science? And if yes, from which subdiscipline? Which theoretical debate or strand of literature does the author aims to contribute to? The theoretical perspective would as well be crucial to know how the research questions were derived
Are there one or several authors? At some points it sais “our” or “we” though there is only one author listed.
Section 2.1. headed “Theoretical Background” is actually not a theory section but a section introducing the case study and the context of it.
Section 2.3 is well-elaborated and provides a good and interesting overview of the political processes.
Section 3.1. discusses the results of the text analysis, this analysis would particularly benefit from theoretical embedment. In my opinion, the author tries to work out frames or policy narratives, so a connection to the framing literature (e.g. Scheufele or Tewksbury or Weaver) and method of analysis could be an option. An alternative option would be the connection to the literature on narrative in public policy (e.g. Stone or Shanahan,…). Overall, I think as by now, the major weakness of the manuscript is on the theory side, whereas the empirical part is nicely elaborated but would still benefit from a systematization by means of theoretical embedment.
Author Response
Reviewer 3
Point 1:
What are the possible conclusions or lessons for other regions from this single case study?
Did the author consider the initiative of the EU on illegal dog and cat breeding? How do these come into play? (https://www.europarl.europa.eu/news/en/press-room/20200206IPR72016/stop-illegal-trade-in-cats-and-dogs-says-european-parliament)
Response 1:
I added a reference to EU regulation and in the end I try to generalize the findings from the case study (Conclusion).
Point 2:
The manuscript includes several normative assumptions, these should be eliminated or at least be supported by some literature, e.g. page 1 lines 41/42, but also in the remaining part of the manuscript. In addition, at some points there is a rather colloquial wording (e.g. p. 2 lines 49), which should be revised. Also, I would recommend proof-reading by a native speaker.
Response 2:
I added a source (line 41/42) and changed the formula (line 49). I tried to remove the text from colloquial expressions.
Point 3:
What is the theoretical and disciplinary background? On page 2 it is states that “a legal and political” one, is this political science? And if yes, from which subdiscipline? Which theoretical debate or strand of literature does the author aims to contribute to? The theoretical perspective would as well be crucial to know how the research questions were derived
Response 3:
I added the methodological part (with focus on Qualitative Content Analysis) focusing on the legal analysis of legislation (63-72).
Point 4:
Are there one or several authors? At some points it sais “our” or “we” though there is only one author listed.
Response 4:
I changed “we” to “I”. Sorry for that – this is typical for the Czech texts, that “we” is used.
Point 5:
Section 2.1. headed “Theoretical Background” is actually not a theory section but a section introducing the case study and the context of it.
Response 5:
I changed the head of section 2.1 in Theoretical Background and Case Study.
I believe that could be acceptable in this way.
Point 6:
Section 3.1. discusses the results of the text analysis, this analysis would particularly benefit from theoretical embedment. In my opinion, the author tries to work out frames or policy narratives, so a connection to the framing literature (e.g. Scheufele or Tewksbury or Weaver) and method of analysis could be an option. An alternative option would be the connection to the literature on narrative in public policy (e.g. Stone or Shanahan,…). Overall, I think as by now, the major weakness of the manuscript is on the theory side, whereas the empirical part is nicely elaborated but would still benefit from a systematization by means of theoretical embedment.
Response 6:
I added the study from Scheufele and the Ref. 6 in the methodological note.

Reviewer 4 Report
In this paper, the Author discusses about the kind of main legislative issues of and obstacles there are to providing a legal solution to the problem of illegal puppy mills in the post-communist context
The Author has investigated an interesting and novel topic, and the theme has been properly described.
I would like to congratulate Author for the good-quality of the article and for the clear and appropriate structure. The manuscript is well written, presented and discussed.
In general, the structure of the article is satisfactory and in agreement with the journal instructions for authors.
The objectives of the paper are of interest and fit well within the scope of the journal.
The language is fluent and appropriate.
In my opinion, the manuscript could be accepted for publication in Animals.
Author Response
Thank you for your review report!
Round 2
Reviewer 2 Report
Thank you for the revisions. However, there remain a number of issues:
Methodological choices should be cited as well rather than grounded in opinions (line 66 - 67) -
"It seems to me to be more suitable for grasping the topic and developing a more in-depth understanding (given the topic’s timely as well as problematic nature)". This is inappropriate for an academic article. There are numerous sources to support why content analysis is a justified method.
The structure needs to be reconsidered please. The last part of the introduction is the methodology, which belongs in section two. Here also lies the main improvement for the paper is that it is still not clear why this is important and why your approach of looking at the debate in opinions is relevant. This is hinted at in 2.1.1, but needs to be set out in the introduction. The theoretical background section still is not theoretical, but simply a literature review. This needs to be reconsidered and it links directly to why is this topic important.
The article is still filled with emotive language, which is understandable, but this is not the place for it.
Whilst literature has been added, there are still places where the article is assertive rather than substantiated. This remains around claims about post-communist societies and how we know or don't know about the puppy trade.
Author Response
Dear reviewer,
many thanks for your critical eye. Now I am only responding to your critical points. The text has undergone complete linguistic proofreading from native speaker.
1) I removed the sentence in the lines 66-67. I apologize for this, although I mentioned in the text that I consider (yes - that is my opinion) the relevant choice of the method of content analysis.
2) I added sources to support why content analysis is a justified method (see FN 6).
3) I changed the structure of the study according to your instructions. The way you suggest, is more logical!
4) My biggest problem: why is this topic important. I think that is mentioned in the text. However, I explicitly added to the paragraph (from line 63) that such an intense parliamentary debate on animal cruelty has not yet taken place in the Central European area. That is why this text pays attention to this.
5) I acknowledge here that I used expressive language in describing of this topic. After critique from you and other 3 reviewers, I revised the text once more (and I discussed it with the native speaker). Once again, my colleagues and I went through the text. I want to believe that it is now for you acceptable.
6) I added the footnotes 11, 14 and 18, which respond to your criticism. One source points to the specifics of legal regulations in post-communist countries, the other reinforce my arguments regarding sociel sciences approaches and animal protection.
Best regards
L.

Reviewer 3 Report
Thank you for your answer and the revisions, though the changes in my opinion are only minor. Most likely this is due to the short time frame for reworking the article.
I would have liked to see some further theoretical advancement, which I believe still is a shortcoming of the study. However, I think the strength of the study is the case study which partly compensates for this shortcoming.
Author Response
Dear reviewer,
the text has undergone another proofreading from the native speaker.
I removed emotional language and added some other sources (see e.g. footnotes 11, 14 and 18) about content analysis and about specifics methodological access to the topics of animal protection. I changed the structure of the study, the methodological part is now included in the part 2.
I really focus mainly on this case study. Therefore, I would be very happy if the text no longer had to be theoretically grounded. :-)
Best regards,
L.
